# Genetic Hierarchy of Acute Myeloid Leukemia: From Clonal Hematopoiesis to Molecular Residual Disease

**DOI:** 10.3390/ijms19123850

**Published:** 2018-12-03

**Authors:** Jean-Alain Martignoles, François Delhommeau, Pierre Hirsch

**Affiliations:** Sorbonne Université, Inserm, Centre de Recherche Saint-Antoine, CRSA, AP-HP, Hôpital Saint-Antoine, Hématologie Biologique, F-75012 Paris, France; jamartignoles@hotmail.com (J.-A.M.); pierre.hirsch@aphp.fr (P.H.)

**Keywords:** acute myeloid leukemia, genetic hierarchy, clonal hematopoiesis, molecular residual disease

## Abstract

Recent advances in the field of cancer genome analysis revolutionized the picture we have of acute myeloid leukemia (AML). Pan-genomic studies, using either single nucleotide polymorphism arrays or whole genome/exome next generation sequencing, uncovered alterations in dozens of new genes or pathways, intimately connected with the development of leukemia. From a simple two-hit model in the late nineties, we are now building clonal stories that involve multiple unexpected cellular functions, leading to full-blown AML. In this review, we will address several seminal concepts that result from these new findings. We will describe the genetic landscape of AML, the association and order of events that define multiple sub-entities, both in terms of pathogenesis and in terms of clinical practice. Finally, we will discuss the use of this knowledge in the settings of new strategies for the evaluation of measurable residual diseases (MRD), using clone-specific multiple molecular targets.

## 1. AML Classifications

From a clinical ontogeny point of view, three different kinds of acute myeloid leukemia (AML) can be defined as follows: *de novo* AML, the occurrence of which cannot be linked to a previously-known hematologic disorder, secondary AML (s-AML), which occurs in a context of myelodysplastic syndrome (MDS) or myeloproliferative neoplasms (MPN), and therapy-related AML (t-AML), which develops secondary to cytotoxic and radiation therapies. This clinical classification does not reflect the high heterogeneity of AML, as each clinical category encompasses many different types of diseases with distinct prognosis and molecular features.

The 2016 WHO classification [1] of myeloid neoplasms and acute leukemia defines six subtypes of AML and related neoplasms—AML with recurrent genetic abnormalities, AML with myelodysplasia-related changes, therapy-related myeloid neoplasms, i.e., t-AML and therapy-related MDS (t-MDS)-, AML, not otherwise specified, myeloid sarcoma, and myeloid proliferations of Down syndrome. This classification describes the diagnostic features, as well as the prognostic factors of myeloid malignancies. The WHO classification mainly focuses on molecular, genetic, and cytogenetic alterations that have prognostic values, as each AML subset within the six categories tends to correspond to an idiosyncratic disease evolution. However, behind these categories, underlie complex and variegated genetic interactions and ontogenies that are not properly set-out by the current classifications.

## 2. Genetic Landscape of AML

The WHO classification [1] and large studies [2,3] of AML cases converge to a robust definition of the genetic AML groups, using two types of lesions—recurrent chromosomal changes and gene mutations. The principal lesions of AML, the functional groups to which they belong, their prognostic value, and their eventual associations in a large cohort [2], are listed in Table 1. The most frequent translocations in AML are individualized in specific subgroups, by the WHO classification. These subgroups include acute promyelocytic leukemia with t(15;17) *(PML-RARA*), AML with t(8;21)(q22;q22.1) (*RUNX1-RUNX1T1*), AML with inv(16)(p13.1q22) or t(16;16)(p13.1;q22) (*CBFB-MYH11*), AML with t(9;11)(p21.3;q23.3) (*MLLT3-KMT2A*), AML with t(6;9)(p23;q34.1) (*DEK-NUP214*), AML with inv(3)(q21.3q26.2) or t(3;3)(q21.3;q26.2) (*RPN1-MECOM*), and AML with t(1;22)(p13.3;q13.3) (*RBM15-MKL1*). They were clearly characterized in a large cohort as having a strong prognostic value. On the other side, some gene mutations, such as *RUNX1*, *CEBPA*, and *NPM1* define specific subgroups. All these subgroups and variegated genetic landscapes were unraveled over the past 10 years, thanks to the development of high throughput sequencing technologies, which allowed the characterization of the most frequently-mutated genes in AML [2,4,5]. The top three driver mutations are those in *FLT3*, *NPM1*, and *DNMT3A*, which are especially found in *de novo* AML with normal karyotype. Some other genes, such as *SRSF2*, *SF3B1*, *U2AF1* or *BCOR* are often found mutated in s-AML. Mutations in *TP53*, almost always associated with a complex karyotype, are mainly found in t-AML. Some mutations, including those in *DNMT3A*, *TET2*, and *ASXL1*, have been shown to appear years before the occurrence of AML [6]. They belong to what is called “clonal hematopoiesis of indeterminate potential (CHIP)” because they occur in healthy people, almost always aged >50 years, their incidence increases with age, and they are associated with a higher risk of developing blood malignant diseases [7]. They have been shown to persist at relapse and complete remission (CR) [8,9] and to provide a multilineage selective advantage to the mutant over the normal hematopoietic stem and progenitor cells (HSPCs), in vivo [8,10,11]. They are, therefore, considered as preleukemic events which initiate the disease. In most cases of AML, mutations are not found in isolation but are either associated with other recurrent mutations or with cytogenetic abnormalities. This highlights the need of cooperation between different types of mutations, for the development of an overt AML. Additionally, it asks the fundamental questions of the most frequent associations of mutations, their order of occurrence, and their functional cooperation leading to the dysregulation of normal proliferation and differentiation of the mutated clone, which will eventually give rise to an AML.

## 3. Association of genetic lesions in AML

A large study of one thousand five hundred and forty patients [2] with AML, showed the importance of genes interaction on survival. For instance, the prognostic value of *FLT3^ITD^* mutations in AML changes according to the presence or the absence of co-occurring *DNMT3A* and *NMP1* mutations. Kaplan-Meier curves showed that *FLT3^ITD^* mutations have a statistically-significant bad impact on the overall survival (OS), only if *DNMT3A* and *NPM1*, are both mutated. However, when neither *DNMT3A* nor *NPM1* are mutated, or when only one of them is mutated, the occurrence of an *FLT3^ITD^* mutation does not alter the prognosis. Conversely, when *NRAS^G12/13^* mutations co-occur with *DNMT3A* and *NPM1* mutations, the OS is statistically improved. Then again, if neither *DNMT3A* nor *NPM1* are mutated or if only one of them is mutated, the occurrence of a *NRAS^G12/13^* mutation does not change the prognosis. These results reveal the complex network of gene-gene interaction as prognostic modifiers. Moreover, it is tempting to consider that these specific interactions confer particular functionalities to the mutated clone.

Along with the recurrent translocations, which have specific functional consequences, genetic changes in AML can be categorized into functional groupings [5,12,13], according to their function on DNA methylation, chromatin modifications, RNA splicing, the cohesin complex, *NPM1*, transcription factors, tumor suppressors, and the signaling pathways (Figure 1A). The relevance of this functional categorization has become clearer in these last few years by the assessment of the clonal hierarchy of AML, which has unraveled recurrent schemes in the association and the order of mutations [2,3,4,5,11]. As a matter of fact, the order of occurrence of mutations within each functional group is not random, as it appears that mutations belonging to the DNA methylation group, such as *DNMT3A* or *TET2*, or to the chromatin modifications group, such as *ASXL1*, often appear first in clonal evolution. Conversely, mutations in *NPM1* or in proliferation signaling pathways, such as *FLT3*, *KIT*, *NRAS* or *KRAS*, mostly occur later or even as the last event in the clonal phylogeny. The type of association and the order of mutations have an impact on the kinetics of the development of AML. It is believed that some associations of mutations will initiate a leukemia, in a very long period of time, up to a few decades, but others will engender a more powerful proliferative advantage to the mutated clone, leading to a faster occurrence of AML. The kinetics of development strongly depends on the functional type of the first event initiating the disease, its synergistic effect on the second, and other successive mutations for clonal expansion. It depends also on the moment of appearance of the mutation during the hematopoietic development; the consequences of the mutation cannot be the same if it occurs in a hematopoietic stem cell, which is quiescent and enters rarely into a division cycle, or if it appears in a highly proliferative progenitor.

## 4. Distinct Genetic Hierarchies of AMLs

Genetic hierarchies of AMLs can be defined and delineated, depending on associations and sequences of appearance of genetic lesions. In several works, efforts were made to conciliate the clinical and genetic classifications. Altogether, they identify the genetic hierarchies specific for *de novo* AMLs, genetic hierarchies specific for AMLs with myelodysplastic features, or characteristic of secondary AMLs, and genetic hierarchies typical of *TP53*-mutated neoplasms [2,3,11]. These three classes of genetic hierarchies will, therefore, be named “*de novo*-type AMLs”, “secondary-type AMLs”, and TP53-type AMLs, further in this review.

### 4.1. De novo-Type AMLs

Among *de novo*-type AMLs, one can distinguish several types of genetic hierarchies (Figure 1B). The first category corresponds to the successive occurrence of a mutation in a gene coding for epigenetic regulation, including DNA methylation (*DNMT3A* and *TET2*), followed by a mutation in *NPM1* or in a classically-mutated transcription factor in AML (*RUNX1*, *CEBPA*, and *GATA2*), then the last mutation arises in a gene involved in the signaling pathways (*FLT3*, RAS, *KIT* or other TK/RAS pathway genes) [11]. Single-cell based analyses, as well as clonal reconstructions, using large deep sequencing datasets, indicated that the order of lesions was invariably the same in this AML category [2,3,4,11]. Of note, almost 90% of the AMLs following this type of clonal hierarchy are *de novo* AMLs with normal karyotype, of which *DNMT3A* and *NPM1* mutations represent the most classical association. In the cohort of one thousand five hundred and forty AML analyzed by Papaemmanuil and colleagues [2], among those mutated for *NPM1*, the most frequently-associated mutated genes were CHIP mutations—*DNMT3A* which was found mutated 54% of the time and *TET2* in 16% of cases—, and mutations in tyrosine-kinase and RAS pathways genes, which were found in 39% of cases for *FLT3*, in 19% for *NRAS*, and 15% for *PTPN11*. Through variant allele frequency (VAF) analysis, the order of mutations showed that CHIP mutations appeared first, followed by *NPM1*, and then by mutations in signaling pathway genes. These data established this kind of clonal hierarchy as the archetype of CHIP-derived AML. Taken together, *NPM1* mutations were associated, in 75% of cases, with one or more CHIP mutations [13], or *IDH1* and *IDH2* mutations, which are known to be mutually exclusive with *TET2* and *WT1*, because of their similar consequences on DNA hydroxymethylation and similar role in leukemogenesis [14,15,16]. Conversely, IDH^R172^ mutations, the least frequent mutations of *IDH2*, have been shown to be mutually exclusive with *NPM1* [2], suggesting a different leukemic development for this specific mutation.

The second type of clonal hierarchy in *de novo*-type AMLs follows almost the same pattern as the first group but lacks the CHIP-associated mutations. Indeed, the first mutation occurs in *NPM1* or in transcriptions factors (*RUNX1*, *CEBPA*, and *GATA2)*, followed by mutation in genes involved in proliferative pathways [11]. This is particularly true for the bi-allelic mutation of *CEBPA* (*CEBPA*^bi-allelic^), which is not associated with mutations in CHIP-related genes, in more than 80% of cases, but is associated with *NRAS* mutations in 30% of cases [2] or with mutations in *GATA2* (20–30%), *WT1* (20%), and *CSF3R* (20%) [2,13,17]. *CEBPA*^bi-allelic^ is a recognized AML subgroup by the 2016 WHO classification of myeloid neoplasms and acute leukemia. From a development point of view, *CEBPA*^bi-allelic^ must occur in later stages, during the hematopoietic development, as opposed to the CHIP mutations which occur in early hematopoietic stem cells (HSC), because *CEBPA* plays a fundamental role in the maintenance of HSC pool and in the priming of myeloid differentiation [18,19,20]. Several *Cebpa*-knockout mouse models [20,21] show that early defect in *Cebpa* leads to an exhaustion of HSC pool and abortion of myeloid commitment. As opposed to *CEBPA*^bi-allelic^, mono-allelic mutations of *CEBPA* (*CEBPA*^mono-allelic^) rarely occur alone in AML and are often found associated with *NPM1*, *FLT3*, or *ASXL1* mutations [22,23]. These data indicate different possible cooperation pathways for a *CEBPA*^mono-allelic^ mutation to drive AML, by either cooperating with mutations involving genes from other functional subgroups (CHIP, *NPM1*, signaling pathway) or with a second mutation in *CEBPA*. As the first category, this type of clonal hierarchy is mainly found in *de novo* AML with normal karyotype.

The third clonal hierarchy encountered in *de novo*-type AML is designated by specific chromosomal aberrations as the initiating event, often followed by mutations in tyrosine-kinase or RAS pathways, but never associated with the *NPM1*, *CEBPA*, *RUNX1*, or *GATA2* mutations. These chromosomal aberrations correspond to the recurrent rearrangements included in the WHO category “AML with recurrent genetic abnormalities”, such as t(8;21)(q22;q22.1), inv(16)(p13.1q22) or t(16;16)(p13.1;q22), and *MLL* (also known as *KMT2A*) fusion genes. Amongst the two hundred AMLs of the cancer genome atlas [5] cohort, none of the twenty-four cases of the t(8;21), inv(16), or the MLL fusion genes, are associated with either *NPM1*, *CEBPA*, or *RUNX1* mutations. By contrast, they are frequently associated with *FLT3*, *KIT*, or *RAS* mutations. The very large cohort of Papaemmanuil et al. [2], also finds that 23% of AMLs with MLL fusion genes, present NRAS mutations. Concerning t(8;21)- and inv(16)-type AMLs, the same large cohort confirms the absence of association with mutations in *NPM1* or in the previously cited transcription factor genes, and shows an association between t(8;21) and *KIT* in 38% of cases. This association has been shown to have poorer outcomes [24]. An even stronger association between inv(16) and NRAS mutations is found in 53% of cases. Using single-cell-derived colony genotyping and FISH (fluorescence *in situ* hybridization), we analyzed the clonal hierarchy of four AMLs with MLL fusion genes [11]. For each AML, the chromosomal aberration appeared first, and the latest mutation always involved a proliferative signaling pathway gene. These results are in line with other studies showing that chromosomal aberrations mentioned above are initiating events, as they occur early in the clonal history, or can lead to pre-leukemic hematopoiesis in xenotransplantation models [25,26,27].

### 4.2. Secondary-Type AMLs

#### 4.2.1. Secondary-Type AMLs to Myelodysplastic Syndromes

In secondary AML (s-AML), which is a particularly frequent complication of myelodysplastic syndrome, as more than 20% of MDS transform into s-AML [28], the association of mutations differs from *de novo* AML (Figure 1C). Lindsley [3] and collaborators genotyped a cohort of one hundred and ninety-four AMLs, composed of s-AML and t-AML, ninety-three of those were secondary to MDS or to chronic myelomonocytic leukemia (CMML). Eight genes involved in three functional groups were found to be specifically associated with s-AML: RNA splicing for *SRSF2*, *U2AF1*, *ZRSR2* and *SF3B1*, chromatin modifiers for *BCOR*, *ASXL1*, and *EZH2*, and cohesin complex for *STAG2*. The most frequently-mutated genes are *ASXL1* (32%) and *SRSF2* (20%), which also correspond to the most frequent association, between the eight genes. Indeed, 63% of *SRSF2* mutated s-AML are also mutated for *ASXL1*. The latter is also frequently associated with *EZH2*, *ZRSR2*, *STAG2*, and *U2AF1*. Being one of the most commonly CHIP-associated gene [6], *ASXL1*-repeated association with the other genes makes sense from a statistical point of view, but also, from a clinical perspective, because MDS principally occurs in elderly patients who are mostly affected by CHIP. DNA methylation-associated genes, such as *TET2* and *DNMT3A*, extremely frequent CHIP-associated genes, or *IDH1/2*, are found to be mutated in forty-four out of the ninety-three s-AML in this cohort, and are more than half the time, associated with one of the eight s-AML specific mutations. The second overall most frequently mutated gene in this s-AML cohort is *RUNX1*. Out of the ninety-three s-AML, twenty-nine (31%) carry *RUNX1* mutations and twenty-five co-occur with one or more of the eight specific mutations, especially with *ASXL1* and *SRSF2*. *RUNX1* cannot be categorized as s-AML-specific mutated gene because of its high prevalence in *de novo* AML. Likewise, signaling pathway genes are also mutated in more than 50% of s-AML.

To assess the question of order of appearance of mutations, one must look backwards and investigate mutational profiles of MDS. Haferlach et al. [29] analyzed the mutational profiles of nine hundred and forty-four patients with MDS and revealed that the most frequently-mutated genes belonged to the RNA splicing functional group, found mutated in 64% of cases, followed by mutations in DNA methylation-associated genes (50%), in chromatin modifiers genes (30%), in transcription factor genes (30%), then in signal transduction pathways (15%), and cohesin complex genes (15%), and finally mutations in the DNA repair (10%). Six genes were found mutated in more than 10% of the cases: *TET2*, *SF3B1*, *ASXL1*, *SRSF2*, *DNMT3A*, and *RUNX1*. Other recurrently mutated genes (found mutated in 2–10% of cases) were *U2AF1*, *ZRSR2*, *STAG2*, *TP53*, *EZH2*, *CBL*, *JAK2*, *BCOR*, *IDH2*, *NRAS*, *MPL*, *NF1*, *ATM*, *IDH1*, *KRAS*, *PHF6*, *BRCC3*, *ETV6*, and *LAMB4*. The same mutational landscape was observed by numerous other smaller scale studies [30,31,32,33,34,35]. Among these genes, the most recurrent mutations associations were between *STAG2*, *IDH2*, *ASXL1*, *RUNX1*, and *BCOR*. *ASXL1* mutations were the most frequent associated mutations. The *SF3B1* and *DNMT3A* mutations were less likely associated with *ASXL1*. *TET2* mutations showed positive correlations with *SRSF2* and *ZRSR2* mutations. The *TET2/SRSF2* association was also pointed in the context of CMML [36]. Interestingly, a mutually exclusive relationship was found between *SRSF2* mutations and those in the *DNMT3A* and *EZH2*. Regarding *SF3B1* mutations, they were less associated to the other lesions, apart from *DNMT3A* and *JAK2*, and occurred mainly in MDS with ring sideroblasts (MDS-RS). The link between *SF3B1* mutations and MDS-RS was confirmed later on by another large-scale study, showing that *SF3B1* mutations occurred in 83% of MDS-RS cases [37]. The significant association between *SF3B1* and *DNMT3A* mutations was confirmed more recently [31]. Concerning the order of appearance, the VAF analysis showed that mutations in genes involved in DNA methylation and spliceosome appeared before all the others and that mutations in tyrosine kinase and RAS pathways occurred last. These results agree with many other studies [28,30,32,34,36,38]. Thus, the order of appearance of mutations in s-AML can mainly be described as follows: Mutations in DNA methylation genes, RNA splicing, and chromatin modifiers first appear, followed by mutations in transcription factors, and finally in signaling pathways. Of note, mutations in the tyrosine-kinase and RAS pathways are predictive of evolution into s-AML [3,28,32]. Concerning the cytogenetic abnormalities (e.g., monosomy 7, trisomy 8, del(5q), and del(20q)), even though they were shown to mostly occur as second or later events, they could also occur early in some cases or they can constitute a minor independent clone [28,34].

#### 4.2.2. Secondary-Type AMLs to Myeloproliferative Neoplasms

BCR-ABL-negative myeloproliferative neoplasms (MPN), represented by polycythaemia vera (PV), essential thrombocythemia (ET), and primary myelofibrosis (PMF), can evolve during the course of the natural history of the disease to what is called blast-phase, which is namely, s-AML to MPN (Figure 1C). The incidence of s-AML can vary from less than 1% for ET, around 7% for PV, and more than 20% for PMF [39,40]. The three classical driver mutations in *JAK2*, *CALR*, and *MPL*, are the unique lesions found in 45% of patients, but in the remaining 55% other mutations can be found, during the chronic phase of the disease. A 2018 study by Grinfeld et al. [41] analyzed a broad cohort of two thousand and thirty-five patients suffering from chronic-phase MPN. The most common genetic alterations, which occurred in 5% to more than 15% of cases, were chromosome 9 LOH (loss of heterozygosity), causing *JAK2* homozygosity, and mutations in the three most frequently encountered CHIP genes, *TET2*, *ASXL1*, and *DNMT3A*. Alterations in *EZH2*, *IDH1*, *IDH2*, *PHF6*, *CUX1*, *ZRSR2*, *SRSF2*, *U2AF1*, *KRAS*, *NRAS*, *GNAS*, *CBL*, *RUNX1*, *STAG2*, *BCOR*, *PPM1D*, *TP53*, and chr4q and chr7/7q LOH were less frequent, as each of them account for less than 5% of cases. Concerning the order of occurrence, *CALR* and *MPL* are usually the first event, but for *JAK2* mutations, this study confirmed the previous results [42,43] showing variegated patterns. *JAK2* mutations were shown to appear first, followed by mutations in chromatin modifiers, such as *ASXL1* or *EZH2*, especially in the PMF cases. Conversely, when *JAK2* appeared to be sub-clonal to *TET2*, *DNMT3A* or *SF3B1* mutations, it occurred in the ET cases. These data highlight the fact that the order of occurrence of mutations has an impact, as well as the type of mutations in the pathophysiology and, thereby, in the phenotype induction. The risk of leukemic and myelofibrotic transformation, in line with other studies [44], was associated with mutations in epigenetic regulators, such as *IDH1/2*, chromatin modifier (*EZH2* and *ASXL1*) splicing factors (*SRSF2*), RAS signaling, and in *TP53*, which were more likely to appear as second or later event, in the course of the disease. In order to identify mutations in the blast-phase, compared to the chronic-phase MPN, Lasho et al. [45] analyzed the mutational landscapes of seventy-five AMLs, secondary to MPN, and nineteen paired chronic-blast phase samples. The most frequent acquired pathogenic mutations were in *ASXL1* (47%), *TET2* (19%), *RUNX1* (17%), *TP53* (16%), *EZH2*(15%), *SRSF2* (13%), and *IDH1* (12%). A relative mutual exclusivity for *TP53*, *NRAS*/*KRAS*, *EZH2*, *SH2B3*, *RUNX1*, and *SRSF2* was observed. Interestingly, associations were seen between *CEBPA* and *MPL*, and between *NRAS* and triple-negative diseases (i.e., no mutation in *JAK2*, *MPL*, and *CALR*). Nineteen cases of paired chronic-blast phase samples were analyzed. The following mutations have appeared in the blast-phase: *EZH2* and *PTPN11* in 67% of cases, *SH2B3* in 60%, *TET2* in 50%, and *ASXL1* in 33%. Two cases of *TP53* mutations were only found in the blast-phase samples. These data confirmed the results of a previous study, which reported increased prevalence of the *TP53* (27%), *PTPN11* (7%), *ASXL1* (47%), *IDH2* (31%), and *SRSF2* (22%) mutations in blast-phase [46]. It is worth noting that the incidence of *TP53* mutations is increased in s-AML, as compared to *de novo* AML, this partially explains the dismal prognosis of s-AML.

### 4.3. TP53-Type AMLs

*TP53* mutated AML account for 12–13% of all AML cases, they are associated with complex karyotype and display particular dismal outcomes [2,47,48]. Their incidence is higher in t-AML subset, by up to 23% [3]. This higher incidence of *TP53* mutations in t-AML has recently been demonstrated by Wong and collaborators [49], who showed that cytotoxic chemotherapy selects pre-existing HSC, mutated for *TP53*. These pre-existing *TP53* mutations correspond to CHIP and are resistant to chemotherapy or radiation regimen, thereby, giving a selective advantage to the mutated-clone over the normal chemo-sensitive HSC, which could expand after treatment. Of note, no difference between chemotherapy and radiation regimen concerning the development of t-AML, has ever been shown. T-AML represent around 7% of all AML cases [50] and those which do not present mutations in *TP53*, either display *de novo*-type mutation association or s-AML-type pattern of mutations [3]. Papaemmanuil et al. [2] established a new AML sub-entity called TP53-aneuploidy, which encompasses *TP53* mutations, complex karyotype, and chromosomal copy-number alterations, such as −5/5q, −7/7q, −17/17p, −12/12p, +8/8q for the most frequent and the most repeatedly associated with *TP53* mutations (Figure 1A–C). Of note, this group presents less RAS-pathway mutations than all the other sub-groups defined by the study, which is interpreted by the authors, as redundancy between RAS activating mutations and a lack of RAS regulators, due to aneuploidy. Mutation in *NPM1*, transcription factors, such as *CEBPA*, *GATA2*, and *RUNX1* were never or seldom associated with the *TP53* mutations. Among the frequent CHIP mutations, *DNMT3A* mutations were the most frequently-associated, even though it remained a rare association.

### 4.4. Clonal Hierarchies in Children AML and Inherited AML Predisposition Syndromes

AML is a rare disease in adults and is even more uncommon in children. All the types of association described in adults also exist in children, but their proportions vary. The genetic hierarchy reminiscent of CHIP, which represents about 40% in adults, accounts for less than 10% in children [2,11,51]. Conversely, the fusion transcripts involving *CBF* and *MLL* rearrangements represent 24% and 21% of all AML cases in patients between 0 and 18 years of age [51]. MLL rearrangements are especially encountered in infants below 2 years (45% of all AML cases) [52], which suggests early events happening during fetal life, as it was previously demonstrated for certain kinds of MLL-rearrangements [53]. The incidence of CBF peaks between 2 and 12 years, they represent 27% of all AML cases in this age subgroup [52].

Among AMLs in children and young adults, there is an increased proportion of cases originating from familial AML predisposition syndromes when compared to adult or elderly AMLs. Several types of inherited forms of AMLs exist, involving multiple predisposition alleles that affect distinct cellular pathways [54]. Most cases originate from the disturbed hematopoietic functions, due to mutations that will affect one or several hematopoietic lineages. Hence, some bone marrow failure syndromes, some isolated inherited cytopenia, eventually associated with immune dysfunction or broader syndromic organ dysfunctions, can evolve into AML. The predisposition alleles involved in these inherited AMLs are those of Fanconi anemia, those affecting telomerase function, those leading to thrombocytopenia (*RUNX1*, *ETV6*, and *ANKRD26*) [54,55,56,57], those involved in congenital neutropenia (*SBDS* and *CSF3R*) [58,59], and *GATA2* [60]. AML occurring in Down Syndrome also belong to this group of diseases [1]. In some conditions, there is no hematopoietic phenotype prior the occurrence of a myeloid neoplasm or an AML. This is the case in *CEBPA* [61], *SRP72* [62], *ATG2B/GSKIP* [63] duplication, and *DDX41* [64] mutant conditions. Finally, some AML occur in the context of a broader predisposition to cancer, for instance in patients with germline *TP53* [65], *BRCA1* or *BRCA2* mutations [54]. One cannot draw a general picture of somatic genetic hierarchies in AMLs occurring in the context of predisposition syndromes. Each condition may lead to a specific clonal history. For example, AMLs in *RUNX1*, *CEBPA*, *DDX41*, or *TP53* predisposition syndromes will frequently involve the alteration of the respective wild-type allele as an important step towards transformation. Conversely, other AMLs can occur after a process where hematopoietic stem and progenitor cells are positively selected as they manage, through acquired mutations, to escape from their inherited dysfunction. One example may be AMLs in the context of Shwachman-Diamond syndrome, where frequent complex karyotype and *TP53* mutations are found [66,67]. 

## 5. From Genetic Hierarchies to Clone-Specific Measurable Residual Disease in AML

The molecular alterations described above, shape the pathophysiology and the prognosis of AMLs. This wide genetic landscape offers us powerful tools to follow the course of the disease after treatment, at a deep molecular level, in order to detect an early relapse and a poor response to therapy, which is not seen by the morphological and immunophenotypical evaluation of the residual disease. Indeed, one can anticipate four general outcomes when one starts an AML treatment (Figure 2). The first outcome is primary refractoriness or very early relapse. In this case, the first round of treatment fails, and complete remission is either not achieved or is achieved for a very short period of time. This condition is easily detected, or predicted, by cyto-morphologic analyses with a 5% sensitivity threshold. The second possible outcome is characterized by the obtention of a good response with a first complete remission, but with a relapse occurring, within a few years after diagnosis. Here, cyto-morphologic examination has no interest, due to its bad sensitivity. Tools able to evaluate in depth the residual disease are needed to predict the eventual relapse. The third outcome is more uncommon, but it corresponds to long-lasting complete remissions with the possible persistence of initiating lesions. In this case, molecular tools are needed to assess the persistence of the initiating lesion, thus, indicating that the patient is back to a “pre-leukemic” state, with a persistent risk to evolve toward a late relapse. Finally, the last outcome is the cure of the disease. It can be obtained using standard chemotherapy protocols in AMLs, from the group of good prognosis, but the use of hematopoietic stem cell transplantation (HSCT) is still the only curative treatment, in most AMLs. The term “cure” is clinical, but it remains of importance to assess that all malignant cells have been cleared from the bone marrow of the patient. Thus, high sensitivity molecular tools remain mandatory to monitor these patients as well.

The ensemble of tools that are used to evaluate the residual disease, after the treatment of leukemia, is called measurable residual disease (MRD). It was previously termed minimal residual disease, because it was defined by the ability to detect diseases below the morphology-based threshold of 5% of blasts in the bone marrow [68]. MRD is currently evaluated in AML patients, using flow cytometry [68,69,70,71,72,73] or targeted qPCR, and seems to be an important tool to refine outcome prediction. Here, we will focus on the molecular MRD evaluation, and how the understanding of AML clonal architecture could impact the development of future standardized tools, for routine evaluation.

Molecular MRD have first been developed and evaluated for fusion transcript quantification, including mainly RUNX1-RUNX1T1, CBFB-MYH11 [74,75,76], and PML-RARA transcripts [77,78] (Figure 3A). Sensitivity of these assays range from 10^−4^ to 10^−5^. According to the different targets and treatments schedules, MRD detection after one, two, or more courses, is associated with a worse outcome (higher CIR (cumulative incidence of relapse) and lower EFS (event-free survival) and OS (overall survival) rates). NPM1 mutations can also be used for the MRD evaluation by allele-specific PCR, with a sensitivity of 10^−5^ (Figure 3). Detection of mutations, even at a low level after one or multiple courses, have also been associated with worse outcome [79,80,81,82]. Taken together, detection of NPM1 mutations and the three main fusion transcripts in CR provide very efficient tools for the prognosis stratification, in almost 50% of cases of adult AML. The other half of patients cannot currently be evaluated with a high sensitivity method, due to a lack of a usable target. The expression of WT1 can be monitored in most patients, as numerous patients harbor a high expression at diagnosis. This has been linked to AML prognosis, but is recommended only in absence of other evaluable target [68,83,84] due to its low sensitivity around 10^-3^. However, with the recent identification of recurrently mutated genes using NGS (next generation sequencing) technologies, it seems that at least one molecular mutation can be identified in more than 90% of adult AML patients [2,5,85]. Indeed, most AML patients harbor multiple mutations (ranging from 5 to 10) that could theoretically be used for MRD evaluation. From the recent knowledge of AML clonal architecture, several questions arise: Are all genes suitable for MRD evaluation or do some mutations frequently persist at a high level after treatment? Is mutation clearance associated with prognosis? In case of multiple mutations, what are the best markers? What technology can be used in daily practice to ensure multiple mutations detection with a good enough sensitivity for prognostic stratification?

Multi target NGS MRD evaluation and ddPCR evaluation have been tested in a few studies and seems to be able to predict prognosis. The first survey using NGS targeted resequencing for MRD analysis, demonstrated in a cohort of 50 patients, that all mutations found at diagnosis can be searched for in post-induction samples [86]. The detection of any initial mutation with a variant allele frequency over 2.5% (i.e., 5% of cells for heterozygous mutations) was significantly associated with a worse short-term prognosis, opening the way for multi-target NGS evaluation.

Other recently published series [87,88,89] have confirmed this finding. Different VAF thresholds were used to define positive MRDs, after the induction course, ranging from 0.5 to 2.5%. Some studies also compared detection or no detection, without any clear cut-off value, which seems more pertinent in theory but is very dependent on the sensitivity of the assay. Taken together, these studies showed that the detection of any mutation, after one course, in patients that reached cytological CR, is independently-associated with a more important risk of relapse and a shorter OS. These differences were more prominent when excluding the three major genes associated with CHIP (i.e., *DNMT3A*, *ASXL1*, and *TET2*) from the analysis. Persistent mutations in these isolated genes was not associated with prognosis. In line with these results, persistence or emergence of *DNMT3A* mutations after chemotherapy, have already been described, and was not associated with short term prognosis [49,90,91], but seem to sign a reinitialization of the clone, from the first event. Some studies identified other genes that could also persist with a high VAF, including *TP53*, *IDH2*, *IDH1*, *STAG2*, and others [92,93,94]. These genes are mostly implicated in CHIP [6] and are, consequently, frequently (but not always) the founder lesions of the clone. The interest of their use for MRD detection has been debated [88,90,92,94], as their detection could only sign a return to CHIP. Their interest for MRD evaluation could be limited to the cases where they are not the founding event.

An evaluation of clone specific MRD could be useful to overcome the problem of choosing the best target (Figure 3). In a recent work, our team evaluated the interest of such a strategy. Using a combined high sensitivity NGS assay (sensitivity of detection of at least 0.002 or lower, according to the different targets) and the detection of persistent cytogenetic abnormality using FISH, in a limited series of patients, it seems that the persistence of only one detectable event was not a very powerful tool, when the detection of the two or three first events of the clone, after one course (i.e., neglecting the first event regardless of the gene or chromosomic event implied), more efficiently discriminates patients with good or bad prognosis. Further work is being carried out to confirm these findings, but it seems that rather than evaluating one gene, an evaluation of the two or three first events of the clone in every patient, is a more powerful strategy. Of note in this work, when the initiating event was a recurrent translocation (CBF leukemia, or *KMT2A/MLL* translocation), all patients were good responders. As specific PCR MRD evaluation is already well-validated for these recurrent translocations, such strategies should be restricted to the other patients. This strategy seems to be useful in all other categories of AML, including *de novo* AML, s-AML, and *TP53* AML, even if multiple molecular or chromosomal events are often detected in CR, in this last category. Technical progress, based on error-corrected NGS (using molecular barcodes) could allow a more sensitive detection of targets [95,96]. However, the usefulness of reaching sensitivity under 10^−3^ or 10^−4^, for multi-target MRD remains not proven, yet.

The use of similar NGS-based strategies at other time points (after multiple courses, before transplantation or as post therapy monitoring) has not been well studied, yet. One recent study [95] evaluated the interest of target resequencing before allogeneic transplantation. Persistence of detectable MRD was associated with lower OS and RFS (relapse-free survival) probabilities. The interest of long-term monitoring after HSCT or chemotherapy, to predict molecular re-evolution, has not correctly been studied yet. Interestingly, in a few cases of very late relapse (more than four years after diagnosis), the clonal composition between diagnosis and relapse samples were slightly different, but the initiating lesion was the same. In a few cases, this event was also detected at a low frequency, during years in CR samples. This suggests a complete re-evolution of the disease from a persistent CHIP clone, after the initial treatment. Persistence of the first event of the clone after the initial treatment might not be a powerful marker for short-term relapse but might identify patients with a risk of long- term re-evolution, with a second disease and a different clonal composition, deriving from the same persisting event. Accumulation of new events from the persistent CHIP clone might take several years. Multiple case of second donor derived disease after allo-HSCT have been reported. Founding mutations of the second disease were identified either in the cells from the graft, or in the donor, many years after the graft. These underlines the potential risk of a return to CHIP [97,98]. As the proportion of patients with late relapse is very low, the usefulness of such a monitoring system must be discussed. No tool seems to be appropriate for such evaluation, as every potential target should be explored with a high sensitivity, which is not possible by now. The interest of a precocious therapeutic intervention, in absence of real target therapy, has not been proven. Taken together, persistence of the first event of the clone after treatment is associated with a higher risk to redevelop a second disease compared to a whole clone clearance. This risk is not quantified but seems to be low. Further studies are mandatory to evaluate how to monitor these patients.

## 6. Conclusions

AML is an exquisitely complex and heterogeneous type of blood cancer. The outstanding progresses of sequencing techniques from the last decades led us to a precise view of how each kind of AML occur in children, young and old adults, in patients with germline defects, or in patients simply predisposed by aging, or by a toxic exposure. All of this new information allows us to refine our practical classifications, which are now relying not only on the prognostic value of each sub-entity, but also on its precise clonal composition. The multiple genetic hierarchies found in AML can be considered as a double signature. On one hand, although we do not fully understand the mechanisms that underlie the development of AMLs, the history of the clone is the signature of the multiple events that made a hematopoietic stem or progenitor cell become a true leukemic cell. On the other hand, each genetic hierarchy is also the signature of the expected outcome of the disease. Some sub-entities are considered to be good prognosis, and others to be bad prognosis. Along with this static picture at the time of diagnosis, one can also predict the AML outcome by using MRD tools, which gives a dynamic view of the clonal evolution. Thanks to the implementation of new molecular tools, we are now able to back-track the AML clone in a comprehensive manner. In the future, we will be able to delineate what are the key lesions that must be evaluated in an MRD setting, to propose unprecedented personalized treatments.

## Figures and Tables

**Figure 1 ijms-19-03850-f001:**
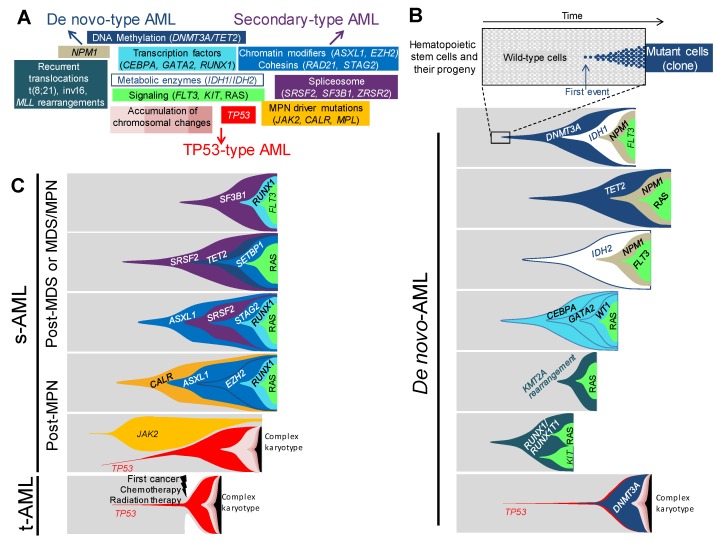
Associations and orders of acquired lesions in *de novo*, secondary, and therapy-related AMLs. (**A**) A schematic view of the multiple lesions underlying leukemogenesis and AML ontology, according to the classification in *de-novo*-type, secondary-type, and TP53-type AMLs—central lesions are shared by all types of AMLs, whereas outlying ones are either specific to or enriched in the type of AML indicated by the nearest arrow. Only the most frequent lesions are indicated in each category. (**B**) Fish diagrams of representative (but not exhaustive) clonal AML hierarchies in *de novo* AMLs. Grey areas show normal hematopoietic stem cells and their progeny. The onset of a clone is achieved through the acquisition of a genetic lesion (first lesion) and the subsequent expansion of mutant cells (colored cells and areas), as indicated in the top panel. Subsequent events will shape the clones with time and lead to AML. Lesions are color-coded, as indicated in panel A. (**C**) Fish diagrams showing representative (but not exhaustive) clonal AML hierarchies in secondary AMLs (s-AML) and therapy-related AMLs (t-AML), as in B.

**Figure 2 ijms-19-03850-f002:**
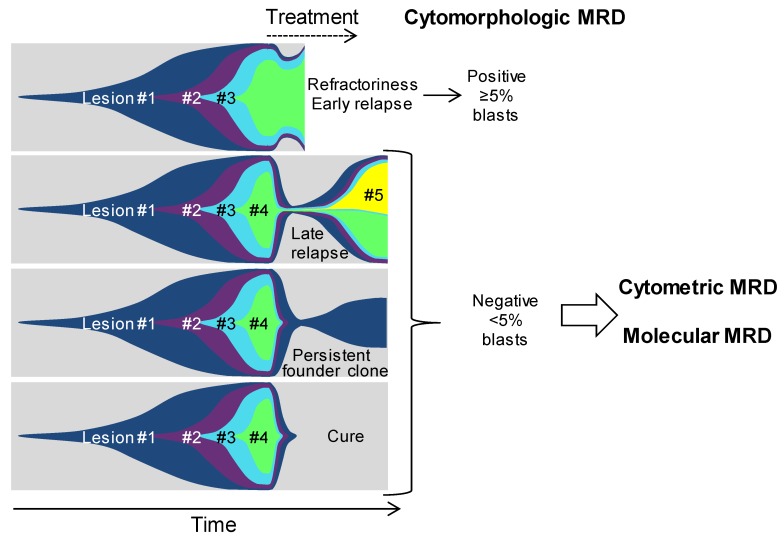
Possible outcomes in AML. Fish diagrams showing clonal evolution in the four possible AML outcomes. Refractoriness and early relapse are diagnosed using cytomorphology and are characterized by the early persistence of most, if not all, lesions, found at the diagnosis. Later relapse occurs after a period of complete remission, where no excess blast is seen, but where most lesions persist at low levels. A peculiar condition is represented by the persistence of clonal hematopoiesis in patients who are in long-lasting remission but retain a founder clone with the most frequent *DNMT3A* mutations. Finally, patients are cured when all mutant cells are cleared from the bone marrow. In these last three outcomes, cytometric and molecular measurable residual diseases (MRD) evaluations are critical for patient monitoring. #1, #2, #3 etc. indicate the successive genetic lesions.

**Figure 3 ijms-19-03850-f003:**
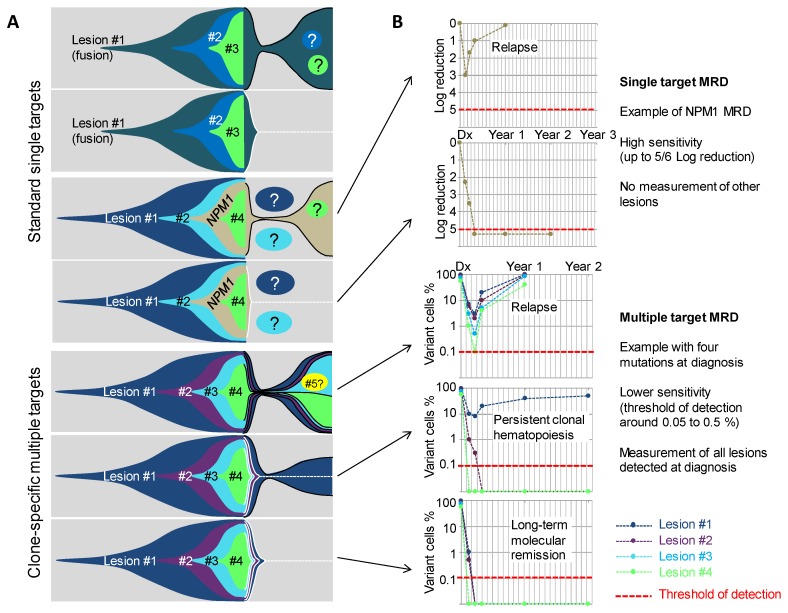
Molecular MRD using single and multiple targets to track AML clones. (**A**) Fish diagrams showing clonal evolution in AMLs with various combinations of lesions and outcomes. MRD Target lesions are indicated by contoured areas in the right part (post-diagnosis) of the diagrams. Black contours indicate lesions that are found MRD-positive, and white contours and dotted lines indicate lesions that become progressively not detectable. Circles with question marks indicate lesions that are not detected by the MRD test. (**B**) Simulations of MRD monitoring of five cases from (A) using a single target MRD test (upper panels) or a clone-specific, multi-target MRD approach (lower panels). Red lines indicate the theoretical sensitivity of the assay. #1, #2, #3 etc. indicate the successive genetic lesions.

**Table 1 ijms-19-03850-t001:** Most frequent cytogenetic lesions and gene mutations in acute myeloid leukemia (AML). The prognostic value of each lesion is indicated, according to the European Leukemia Net groups, when available. Exclusion and associations of lesions were adapted from Papaemmanuil et al. [2].

Category of lesions	Mutated Gene or Cytogenetic Lesion	Encoded Protein	European Leukemia Net Prognostic Group	Pairwise Exclusion of Lesions *	Pairwise Association of Lesions *
DNA methylation	***DNMT3A***	DNA Methyltransferase 3α	NA	***CEBPA*****(monoallelic)**, **complex karyotype**, *KIT*, *ASXL1*, *SRSF2*, del(5q)/−5, recurrent translocations	***NPM1***, ***FLT3***, ***IDH1***, ***IDH2***, *PTPN11*
	***TET2***	Tet Methylcytosine Dioxygenase 2	NA	*IDH2*, *ASXL1*, *SRSF2*, complex karyotype, del(5q)/−5, del(7q)/−7, t(15;17)	***NPM1***, *SRSF2*, *STAG2*, *FLT3*
Metabolic enzymes	***IDH1***	Isocitrate Dehydrogenase (NADP(+)) 1, Cytosolic	NA	*TP53*, recurrent translocations, complex karyotype	***DNMT3A***, ***NPM1***, *PTPN11*
	***IDH2***	Isocitrate Dehydrogenase (NADP(+)) 2, Mitochondrial	NA	*TP53*, *NRAS*, *FLT3*-ITD, recurrent translocations, complex karyotype	***DNMT3A****, SRSF2*, *NPM1*
Nucleophosmin 1	***NPM1***	Nucleophosmin 1	Favorable (without *FLT3*-ITD)	**Abnormal karyotype**, ***TP53***, ***CEBPA*** (**bi-allelic**), ***RUNX1***, ***U2AF1***, ***ASXL1***, *SF3B1*, *EZH2*, *KIT*	***DNMT3A***, ***TET2***, ***IDH1***, ***RAD21***, ***FLT3***, ***PTPN11***, *IDH2*
Transcription factors	***CEBPA***	CCAAT Enhancer Binding Protein α	Favorable (bi-allelic mutation)	***NPM1***, *DNMT3A*, *RUNX1*, *NRAS* (if *CEBPA* mono-allelic) complex karyotype, del(5q)/−5,	***GATA2***, ***WT1***, *STAG2*
	***ETV6***	ETS Variant 6	NA	None	*SF3B1*, *KRAS*, inv(3)
	***GATA2***	GATA Binding Protein 2	NA	None	***CEBPA***, *NRAS*
	***RUNX1***	Runt Related Transcription Factor 1	Adverse	***NPM1***, *CEBPA* (bi-allelic), *TP53*, recurrent translocations	***ASXL1***, ***SRSF2***, *EZH2*, *PHF6*, *STAG2*, *BCOR*
	***WT1***	Wilms Tumor 1	NA	*DNMT3A*, *CEBPA* (mono-allelic), *TP53*, complex karyotype, del(5q)/−5	***CEBPA***, *FLT3*, t(15;17)
Signaling	***CBL***	Cbl Proto-Oncogene	NA	None	*SF3B1*
	***FLT3***	Fms Related Tyrosine Kinase 3	Adverse (if ITD, *NPM1* wild type, and high allelic ratio)	***TP53***, ***NRAS***, **inv(16)**, **del**(**17p**)/**−17**, **del**(**5q**)/**−5**, **del**(**7q**)/**−7**, **complex karyotype**, *KRAS*, *NF1*, *KIT*, *SRSF2*, *ASXL1*, del(20q)/−20, inv(3)	***DNMT3A***, ***NPM1***, **t**(**6;9**), t(15;17), *WT1*
	***KIT***	KIT Proto-Oncogene Receptor Tyrosine Kinase	NA	*NPM1*, *DNMT3A*, *FLT3*-ITD	**t**(**8;21**), inv(16)
	***KRAS***	KRAS Proto-Oncogene, GTPase	NA	*FLT3*-ITD	*ETV6*, *inv(3)*
	***NF1***	Neurofibromin 1	NA	*FLT3*-ITD	None
	***NRAS***	NRAS Proto-Oncogene, GTPase	NA	***FLT3*-ITD**, t(15;17), *IDH2*, *CEBPA* (mono-allelic), *TP53*, complex karyotype	**inv(16)**, *GATA2*
	***PTPN11***	Protein Tyrosine Phosphatase, Non-Receptor Type 11	NA	t(15;17), t(8;21)	***NPM1***, *DNMT3A*, *IDH1*, inv(3), del(7q)/-7
Chromatin/Cohesin	***PHF6***	PHD Finger Protein 6	NA	None	*RUNX1*
	***ASXL1***	Aditionnal Sex Comb Like 1	Adverse	*FLT3*-ITD	***RUNX1***, ***SRSF2***, ***STAG2***, *EZH2*, *U2AF1*
	***BCOR***	BCL6 Corepressor	NA	***FLT3*-ITD**	*SRSF2*, *RUNX1*, inv(3)
	***EZH2***	Enhancer of Zeste 2 Polycomb Repressive Complex 2 Subunit	NA	None	*RUNX1*, *ASXL1*, *STAG2*
	***RAD21***	RAD21 Cohesin Complex Component	NA	None	***NPM1***, t(8;21)
	***STAG2***	Stromal Antigen 2	NA	None	***SRSF2***, ***ASXL1***, *BCOR, RUNX1*, *TET2*, *EZH2, CEBPA*
Spliceosome	***SF3B1***	Splicing Factor 3b Subunit 1	NA	*NPM1*	*ETV6*, *CBL*, inv(3)
	***SRSF2***	Serine And Arginine Rich Splicing Factor 2	NA	*DNMT3A*, inv(16)	***ASXL1***, ***RUNX1***, ***STAG2***, *TET2*, *IDH2*, *BCOR*
	***U2AF1***	U2 Small Nuclear RNA Auxiliary Factor 1	NA	***NPM1***	*ASXL1*, del(20q)/-20
Tumor suppressor	***TP53***	Tumor Protein P53	Adverse	***NPM1***, ***FLT3***, *NRAS*, *WT1*, *IDH1*, *IDH2*, *RUNX1*	**Complex karyotype**, **del**(**5q**)/**−5**, **del**(**7q**)/**−7**, **del**(**20q**)/**−20**
Gene fusions	**t**(**8;21**)	RUNX1-RUNX1T1	Favorable	***NPM1***, *DNMT3A*, *IDH2*, *RUNX1*, *FLT3*-ITD, *PTPN11*, complex karyotype	***KIT***, *RAD21*
	**t**(**15;17**)	PML-RARA	Favorable	***NPM1***, ***DNMT3A***, *TET2*, *IDH2*, *RUNX1*, *PTPN11*, complex karyotype	*WT1*, *FLT3*
	**inv(16)**/**t**(**16;16**)	CBFB-MYH11	Favorable	***NPM1***, ***DNMT3A***, *IDH1*, *IDH2*, *RUNX1*, *SRSF2*, complex karyotype, del(5q)/−5	***NRAS***, *KIT*
	**t**(**6;9**)	DEK-NUP214	Adverse	*NPM1*	***FLT3***
	**inv(3)**	RPN1-MECOM	Adverse	*NPM1*, *FLT3*-ITD	*del(7q)*/*−7*, *ETV6*, *SF3B1*, *BCOR*, *KRAS*, *PTPN11*
	**t**(**9;11**)	MLLT3-KMT2A	Intermediate	***NPM1***, *DNMT3A*	None
Complex karyotype	**Complex karyotype**	NA	Adverse	***NPM1****, **DNMT3A***, ***FLT3***, **recurrent translocations**, *NRAS*, *WT1*, *TET2*, *IDH1*, *IDH2*	***TP53***
Other cytogenetic lesions	**del**(**5q**) **or −5**	NA	Adverse	***NPM1***, ***FLT3***, *DNMT3A*, *TET2*, *IDH2*, *WT1*, *CEBPA* (bi-allelic), inv(16)	***TP53***
	**del**(**7q**) **or −7**	NA	Adverse	***NPM1***, ***FLT3-*****ITD**, *DNMT3A*, *TET2*, *WT1*	***TP53***, **inv(3)**, *PTPN11*
	**abn**(**17p**) **or −17**	NA	Adverse	***NPM1***, ***FLT3-*****ITD**, *DNMT3A*, *TET2*	***TP53***
	**del**(**20q**) **or −20**	NA	Intermediate	***NPM1***, *FLT3*-ITD	***TP53***, *U2AF1*
	**other *KMT2A* translocations**	NA	Adverse (except t(9;11))	***NPM1***	None

* Adapted from Papaemmanuil et al [2]. Bold indicates family-wise error rate <0.05. Otherwise, indicated interactions are those with a false discovery rate <0.1. See [2] for details.

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
