# Peer review of "Genetic Hierarchy of Acute Myeloid Leukemia: From Clonal Hematopoiesis to Molecular Residual Disease"

_ijms, 2018, doi:10.3390/ijms19123850_

Round 1
Reviewer 1 Report
This is timely, interesting and well written review. I have no comments to make.
Author Response
We thank Reviewer 1 for her/his kind comment.
Reviewer 2 Report
In the manuscript «Genetic hierarchy of acute myeloid leukaemia: from clonal hematopoiesis to molecular residual disease» the authors review the state-of-the-art on gene mutations in acute myeloid leukemia (AML). The article is well-written and well-structured, and accompanied with nice figures.
I have only minor comments to the manuscript.
1. The authors list a large number of mutations, of which not all might be familiar as typical for AML to the readers. I therefore advise that the authors add a table where they e.g. sort the mutations in categories (like transcription factors), present the full name of the encoded proteins and show mutations that frequently co-appear or are mutually exclusive. Where available, the prognostic impact of the mutation – as sole mutation to keep it simple – would also be of interest to the reader.
2. The authors mention treatment-related AML as consequence of previous treatments with cytotoxic agents (line 22). However, t-AML can also be secondary to radiation therapy, as is shown in figure 1C. Are there differences in t-AML secondary to chemotherapy as compared to radiation? Could the authors discus on that?
3. The authors also focus on molecular residual disease (MRD). Since the rest of the manuscript is divided into de novo AML, and AML secondary to myeloproliferative diseases and earlier cancer therapy, I wander if there could be different strategies in order to detect MRD in patients? As one usually knows the AML etiology, and different mutations are present in the different AML subtypes, would it be possible to address the issue of MRD according to the AML subtype? Could the authors discus about that?
The remaining comments are mostly of a “cosmetic” nature:
4. Spelling: the authors shift between American and British English – both forms are for instance present in the title. MDPI asks for articles in American English. Please revise.
5. Figures: the text in the figures is added in a program that does not detect English as default language. Thus, many words are marked as incorrectly spelled. Please revise.
Figure 3A: in the first two diagrams it says “fusion”. Should it not be “lesion” like in the rest of the figures?
6. Citations: e.g. line 352. It should read [68-73], rather than [69][70][71][72][73][68].
7. Numbers: The spelling of numbers on page 9 is a little confusing. In line 371, “5-10 mutations” clearly marks an interval, whereas “10-4” (first appearance: line 358) presumingly is meant as power of -4. Please adjust the numberings.
8. Lines 274-276: I do not understand this sentence. The “whether” does not fit here. Please revise.
Author Response
We thank Reviewer 1 and 2 for their kind comments and useful suggestions. Our modifications appear in red in the revised version of our manuscript.
1. The authors list a large number of mutations, of which not all might be familiar as typical for AML to the readers. I therefore advise that the authors add a table where they e.g. sort the mutations in categories (like transcription factors), present the full name of the encoded proteins and show mutations that frequently co-appear or are mutually exclusive. Where available, the prognostic impact of the mutation – as sole mutation to keep it simple – would also be of interest to the reader.
We thank reviewer 2 for this suggestion. We added the requested table. We restricted the data to the most frequent lesions, to the ELN prognostic groups, and to the paper of Papaemmanuil for exclusion/association of lesions to keep it as simple as possible.
2. The authors mention treatment-related AML as consequence of previous treatments with cytotoxic agents (line 22). However, t-AML can also be secondary to radiation therapy, as is shown in figure 1C. Are there differences in t-AML secondary to chemotherapy as compared to radiation? Could the authors discus on that?
We added “radiation therapy” in the sentence. In this context of TP53 mutant AML, we did not find any paper that specifically addresses the differences between t-AML following chemotherapy and those following radiation therapy.
3. The authors also focus on molecular residual disease (MRD). Since the rest of the manuscript is divided into de novo AML, and AML secondary to myeloproliferative diseases and earlier cancer therapy, I wander if there could be different strategies in order to detect MRD in patients? As one usually knows the AML etiology, and different mutations are present in the different AML subtypes, would it be possible to address the issue of MRD according to the AML subtype? Could the authors discus about that?
We added the following sentence to discuss this issue : “This strategy seems to be useful in all other categories of AML including de novo AML, s-AML and TP53 AML, even if multiple molecular or chromosomal events are often detected in CR in this last category”.
The remaining comments are mostly of a “cosmetic” nature:
4. Spelling: the authors shift between American and British English – both forms are for instance present in the title. MDPI asks for articles in American English. Please revise.
Correction has been done
5. Figures: the text in the figures is added in a program that does not detect English as default language. Thus, many words are marked as incorrectly spelled. Please revise.
Figure 3A: in the first two diagrams it says “fusion”. Should it not be “lesion” like in the rest of the figures?
Figures have been changes according to these comments
6. Citations: e.g. line 352. It should read [68-73], rather than [69][70][71][72][73][68].
Correction has been done
7. Numbers: The spelling of numbers on page 9 is a little confusing. In line 371, “5-10 mutations” clearly marks an interval, whereas “10-4” (first appearance: line 358) presumingly is meant as power of -4. Please adjust the numberings.
Correction has been done
8. Lines 274-276: I do not understand this sentence. The “whether” does not fit here. Please revise.
Correction has been done
Professeur François Delhommeau
Chef de service du laboratoire d’hématologie cellulaire
Département d’hématologie biologique du GH HUEP
Hôpital Saint-Antoine
184 rue du faubourg Saint-Antoine
75012 Paris
Secrétariat : 33 1 49 28 22 72
Direct : 33 1 71 97 06 30